# An Analysis of the Association between School Bullying Prevention and Control Measures and Secondary School Students’ Bullying Behavior in Jiangsu Province

**DOI:** 10.3390/bs13110954

**Published:** 2023-11-20

**Authors:** Yong Tian, Jie Yang, Feng Huang, Xiyan Zhang, Xin Wang, Lijun Fan, Wei Du, Hui Xue

**Affiliations:** 1School of Public Health, Southeast University, Nanjing 210009, China; tianyong@seu.edu.cn (Y.T.); fanlijun@seu.edu.cn (L.F.); 220203796@seu.edu.cn (H.X.); 2Department of Child and Adolescent Health Promotion, Jiangsu Provincial Center for Disease Control and Prevention, Nanjing 210009, China; xiyan@alu.fudan.edu.cn (X.Z.); wx1995@connect.hku.hk (X.W.); 3Fujian Provincial Center for Disease Control and Prevention, Fuzhou 350001, China; 220203799@seu.edu.cn

**Keywords:** bullying prevention and control measures in schools, bullying victimization, multiple bullying victimization, secondary school students, China

## Abstract

(1) Background: China released regulations on school bullying prevention and control in 2017; however, current research on school bullying in China focuses on exploring influencing factors and lacks empirical research on the effectiveness of anti-bullying policies in schools. The objective of this study was to use an empirical model to explore the association between bullying prevention and control measures and secondary school students’ bullying victimization and multiple bullying victimization in Chinese schools. (2) Methods: Data were derived from the 2019 Surveillance of Common Diseases and Health Influencing Factors among Students in Jiangsu Province. The school’s bullying prevention and control measures, which was the independent variable, were obtained in the form of a self-report questionnaire and consisted of five measures: the establishment of bullying governance committees, thematic education for students, thematic training for parents, special investigations on bullying, and a bullying disposal process. Bullying victimization and multiple bullying victimization, which was the dependent variable, were obtained through a modified version of the Olweus bullying victimization questionnaire. In order to better explain the differences in the results, this study constructed multilevel logistic regression models to test the association between school bullying prevention and control measures and the rates of bullying victimization and multiple bullying victimization among secondary school students at both the school level and the student level. Meanwhile, this study constructed five models based on the null model by sequentially incorporating demographic variables, physical and mental health variables, lifestyle variables, and bullying prevention and control measures in schools to verify this association. (3) Results: A total of 25,739 students were included in the analysis. The range of bullying victimization rates for students in the different secondary schools in this study was between 6.8% and 37.3%, and the range of multiple bullying victimization rates was between 0.9% and 14.8%. The establishment of bullying disposal procedures was strongly associated with a reduction in bullying victimization (OR = 0.83, 95%CI: 0.71–0.99, *p* < 0.05). Establishing bullying disposal procedures was not significantly associated with multiple bullying victimization rates (OR = 0.89, 95%CI: 0.73–1.09, *p* > 0.05). The establishment of a bullying governance committee, thematic education for students, thematic training for parents, and special surveys on bullying were not significantly associated with bullying victimization rates or multiple bullying victimization rates (all *p* > 0.05). (4) Conclusions: Among the current bullying prevention and control measures for secondary school students in China, the establishment of a bullying disposal process was conducive to reducing the rate of bullying victimization, but it was ineffective in reducing the rate of multiple bullying victimization, and the other preventive and control measures did not achieve the purpose of anti-bullying in schools.

## 1. Introduction

Bullying in secondary schools refers to incidents that occur inside or outside of schools, including secondary schools and secondary vocational schools, involving students. It involves one party (an individual or group) intentionally or maliciously engaging in bullying and insults, using physical, verbal, or cyber means. This behavior may result in physical injuries, property loss, or mental harm to the other party (an individual or a group) [1]. Approximately one-third (32%) of students worldwide have reported experiencing bullying from their peers [2]. The problem of bullying among students poses a significant challenge in China. A survey conducted across 11 provinces in the country revealed that approximately 11% of the surveyed secondary school students had encountered incidents of bullying [3]. Exposure to bullying can result in a multitude of adverse consequences for students, encompassing physical harm, self-inflicted injuries, and diminished academic achievement [4]. Bullying may also give rise to enduring detrimental outcomes for the victim, such as mental health disorders, impaired interpersonal relationships, and engagement in delinquent behaviors [5]. In an effort to mitigate the prevalence of bullying among students, schools have implemented measures aimed at the prevention and control of bullying behavior. The efficacy of various bullying prevention and control measures has yielded diverse outcomes [6]. China released a policy document on the prevention and treatment of bullying and violence among primary and secondary school students in 2017 [7]. However, current research on school bullying in China is still at the stage of exploring influencing factors, and there is a lack of research on the effectiveness of anti-bullying policies. Hence, it becomes imperative to investigate the association between school bullying prevention measures and secondary school students’ bullying victimization and multiple bullying victimization in Chinese schools by means of empirical methodologies to provide evidence for the optimization of school bullying prevention measures.

Research on student bullying in some countries commenced earlier, leading to the development of more mature processes and institutional norms for identifying, reporting, and addressing bullying incidents among students. Furthermore, a comprehensive legal framework has been established on this foundation. For example, the United States has established a relatively complete legal system for the prevention and control of student bullying, utilizing strategies such as federal decentralization, regional legislation, and case precedents [8]. In 2013, the Japanese Diet enacted the School Bullying Prevention and Countermeasures Promotion Act, which, for the first time, put the prevention, handling, and response to student bullying on a legalized track, making it a representative anti-bullying legislation in Japan [9]. Finland has also established an emergency mechanism for combating bullying by placing “Zero Intervention Teams” comprised of highly trained educators and psychological counselors in primary and secondary schools to scrutinize and deal with bullying in a careful manner and to carry out various activities aimed at combating bullying [10]. In addition, countries such as the United Kingdom, Canada, Australia, and New Zealand have also enacted specific legislative measures to prevent and mitigate bullying among students, as well as to uphold school safety [11,12].

Academics from other countries have conducted studies on policies and measures related to the prevention and control of bullying among students. For example, Ginette conducted a comparative analysis of anti-bullying policy incentives and quality in Ontario and Saskatchewan, Canada. This analysis revealed that, within the primary and secondary education context, proactive measures played a pivotal role in preventing and intervening in bullying incidents [12]. The study conducted by Smith et al. demonstrated that schools implementing a greater number of preventive measures experienced a reduction in reported incidents of bullying among students [11]. Hatzenbuehler et al. conducted a survey among victims of bullying in grades 9–12 across 28 districts in the United States to assess the effectiveness of anti-bullying laws in addressing various types of bullying [13]. In a controlled study conducted by Farrington and Ttofi, it was discovered that schools that implemented bullying prevention and control measures exhibited a noteworthy decrease in bullying incidents compared to those that did not. These findings suggest that school-based bullying prevention and control measures could effectively reduce bullying incidents among students [14]. A study conducted in Indonesia demonstrated that the inclusion of moral education in the curriculum, along with the implementation of cultural activities, proved to be an effective measure in preventing bullying [15]. A study conducted in Minnesota, USA, revealed that the integration of diversity education activities in schools contributed to a reduction in the incidence of bullying victimization [16]. A study from Sweden confirmed that improving the school climate was one of the most effective measures to reduce the rate of bullying among students [17].

Although China has not enacted a specific bill designated to tackle student bullying, a range of policies pertaining to preventing and controlling such incidents have been implemented since 2017 that provide recommendations and requirements for various stakeholders, including schools, families, and relevant government departments [7,18]. Among other things, the Programme for Strengthening Comprehensive Management of Bullying among Primary and Secondary School Students provides a clear definition of student bullying. Moreover, it outlines crucial elements for an effective mechanism to coordinate prevention and control strategies, including proactive prevention, lawful handling, and the establishment of a sustainable and efficient framework [19]. Domestic research on bullying prevention and control measures in China has mostly focused on the introduction and study of other countries’ policies and measures related to bullying prevention and control. Zhang Baoshu conducted an analysis of the anti-bullying policy in the United Kingdom, highlighting the significance of legislative measures and the role of the education sector in its effectiveness [20]. A study by Jie Lin and Yachun Chao found that positive intervention-based anti-bullying policies in the United States were effective in preventing and addressing bullying in schools [21]. However, there have been limited case studies on bullying policies in China. For instance, Wang Qiuhua conducted a case study in a county in Fujian Province to examine the challenges and issues encountered during the implementation of student bullying prevention and control policies [22].

Based on the existing research literature, we propose the following research hypotheses:

**Hypothesis** **1.**
*School bullying prevention and control measures were strongly associated with a decrease in the rate of bullying victimization among secondary school students.*


**Hypothesis** **2.**
*School bullying prevention and control measures were strongly associated with a decrease in the rate of multiple bullying victimization among secondary school students.*


In China, there is a lack of empirical research to guide the development and implementation of bullying policies and measures due to the late promulgation of bullying prevention and control policies and the implementation of school prevention and control measures. Therefore, this study aimed to examine the relationship between school-based bullying prevention and control measures and bullying victimization, including multiple bullying victimization, among secondary school students in Jiangsu Province. Findings from this study can provide valuable data support for the rationale and implementation of effective school-based bullying prevention and control policies.

## 2. Materials and Methods

### 2.1. Data Source

Data were derived from the 2019 Surveillance of Common Diseases and Health Influencing Factors of Students in Jiangsu Province. This was a cross-sectional survey implemented by the Jiangsu Provincial Center for Disease Prevention and Control to monitor common diseases and health-influencing factors among students. The sample of this survey covered 169 schools in all 13 prefecture-level cities in Jiangsu Province. In order to ensure that the sample covers both urban and rural areas, this survey took an urban area and an agricultural county (district) in each prefecture-level city separately. Eight schools were selected in each urban area (two primary schools, two middle schools, two high schools, one vocational high school, and one comprehensive university), and five schools (two elementary schools, two middle schools, and one high school) were selected in agriculture-related counties (districts). Monitoring was conducted for grades 4–6 in primary school, middle school, and high school, and grades 1–3 in university, with anonymous, self-administered questionnaires administered to whole classes at each grade level. Thus, the sample of this survey was highly representative. This study focused on a survey of a group of secondary school students. After excluding the primary and university student groups and the missing information data for the independent and dependent variables, a total of 25,379 secondary school students were included in the analyses.

### 2.2. Measures

#### 2.2.1. Measures to Prevent Bullying in Schools

This study focused on obtaining information on the implementation of bullying prevention and control measures in schools through an anonymous self-administered questionnaire. The questions in the questionnaire concerning the prevention and control of bullying in schools covered five main areas, all of which were dichotomous variables, mainly: (1) Establishment of a bullying governance committee. The relevant question in the questionnaire was, “Does the school have a student bullying governance committee?”. The corresponding options were: 1. Yes; 0. No. The Bullying Control Committee consisted of the headmaster, teacher representative, counselor, school employee representative, community worker, parent representative, and an expert from outside the school. In the case of high schools, a student representative was also required. (2) Conducting thematic education for students. The relevant question was, “ Is education on the topic of bullying and violence prevention for students conducted regularly?”. The corresponding options were: 1. Yes; 0. No. (3) Conducting specialized training for parents. The relevant question was, “Are special training sessions for parents on bullying organized on a regular basis?”. The corresponding options were: 1. Yes; 0. No. (4) Conducting a special survey on bullying. The relevant question was, “Whether special surveys on student bullying are conducted regularly”. The corresponding options were: 1. Yes; 0. No. (5) Establishing a bullying disposal process. The relevant question was, “Is there a process in the school rules and regulations for handling student bullying? (including early warning, handling, and intervention)”. The corresponding options were: 1. Yes; 0. No. If the participant answered yes, the school bullying prevention and control measure was implemented, and if the participant answered no, the school bullying prevention and control measure was not implemented.

#### 2.2.2. Bullying Victimization and Multiple Bullying Victimization

The revised Olweus Bullying Victims Questionnaire was used in this study to measure bullying victimization and multiple bullying victimization [23]. Students were surveyed to see if they had experienced bullying in or around the school in the past 30 days, which mainly included the following six types: (1) being made fun of maliciously; (2) being demanded for property; (3) being intentionally excluded from group activities or isolated; (4) being threatened or intimidated; (5) being hit, kicked, pushed, or locked up; (6) being made fun of because of their physical defects or their looks. The options corresponding to these six questions were: 1. never; 2, sometimes; 3, often. Bullying was indicated when participants chose “sometimes” or “often”. A student was defined as a victim of bullying when he/she was subjected to any form of bullying with a frequency of “sometimes” or “often”. The Cronbach’s alpha coefficient for this scale in this study was 0.759, which has good reliability.

There were four main types of student bullying which were physical bullying, verbal bullying, relational bullying, and sexual bullying [24]. Among the more common types of bullying were physical, verbal, and relational bullying. Therefore, this study also categorized the above six types of bullying into three types of bullying victimization, including physical bullying victimization only, verbal bullying victimization only, and relational bullying victimization only. The second and fifth bullying behaviors in the questionnaire were defined as physical bullying victimization only. The first, fourth, and sixth bullying behaviors in the questionnaire were defined as verbal bullying victimization only. The third type of bullying behavior in the questionnaire was defined as relational bullying victimization only. Students who had experienced two or more types of bullying were defined as multiple bullying victimization [25].

#### 2.2.3. Covariates

In reference to the relevant literature on bullying in schools, the covariates included in the study were gender (male or female), age (<15 years old or >= 15 years old), region (urban or rural), family structure (nuclear families or non-nuclear families), mother’s education level (primary school and below, junior or senior high school, university and above), smoking (yes or no), alcohol consumption (yes or no), addictive drug use (yes or no), intake of sugary drinks (yes or no), intake of fruits (yes or no), lack of physical activity (yes or no), television viewing (yes or no), computer use (yes or no), lack of sleep (yes or no), depressive symptoms (yes or no), and overweight or obesity (yes or no) [26,27].

### 2.3. Statistical Analysis

This study used SAS 9.4 software to calculate descriptive statistics (including frequency and percentages) and to construct a multilevel logistic regression model. Multilevel logistic regression models with students as level 1 and schools as level 2 explained the clustering of students in the schools, quantified differences in rates of bullying victimization and multiple bullying victimization between schools, and estimated the strength of the association between rates of bullying victimization and multiple bullying victimization and student and school characteristics [28].

In this study, five separate models were constructed using bullying victimization and multiple bullying victimization as outcome variables. Model 1 was a null model; it did not include any student or school-level independent variables and indicated through school-level random effects whether there were differences between schools in bullying victimization and in rates of multiple bullying victimization. Model 2 added general demographic factors as covariates to Model 1; these covariates included gender, age, region, family structure, and mother’s education. Model 3 added physical and mental health factors to Model 2 as covariates, including depressive symptoms and overweight and obesity. In Model 4, lifestyle behavioral factors were added to Model 3 as covariates, including smoking, alcohol consumption, addictive substance use, intake of sugar-sweetened beverages, intake of fruits, lack of physical activity, television viewing, computer use, and lack of sleep. Model 5 is a full model with school-level factors added to Model 4, including the establishment of a bullying governance committee, thematic education for students, thematic training for parents, specialized bullying investigations, and a bullying disposal process. Differences in bullying victimization rates and multiple bullying victimization rates between schools were estimated from school-level variance. The proportion of total variance in the outcome explained by differences between schools was estimated from the intraclass correlation coefficient (ICC). Quantifying school-to-school differences and describing the median increase in the ratio of bullying victimization and multiple bullying victimization after students moved to a high-risk school was found through the Median Odds Ratio (MOR). The percentage change in variance at the school level after the inclusion of different variables was estimated by the percentage change in variance (PCV). Correlations between student and school characteristics and bullying victimization and multiple bullying victimization were estimated by calculating the ratio of ratios (OR) and 95% confidence intervals (CI) [29]. *P*-values less than 0.05 were considered statistically significant differences.

## 3. Results

### 3.1. Sample Characteristics

A total of 25,379 students were included in the analysis. A total of 85.1% of the schools have set up bullying governance committees, 93.3% have conducted thematic training for students, 84.2% have conducted thematic training for parents, 91.6% have carried out special surveys on bullying, and 66.2% have set up a process for dealing with bullying. A total of 17.9% (*n* = 4544) of students were victims of bullying, of whom 61.1% (*n* = 2777) were boys and 38.9% (*n* = 1767) were girls. The percentage of victims of bullying aged 15 years and over was 42.3% (*n* = 2014). A total of 6.5% (*n* = 1651) of students were victims of multiple bullying, of whom 64.4% (*n* = 1064) were boys and 35.6% (*n* = 587) were girls. The proportion of victims of multiple bullying aged 15 and over was 44.8% (*n* = 740). See Table 1 for details.

### 3.2. Bullying Victimization, Multiple Bullying Victimization in Different Secondary Schools

As can be seen in Figure 1, the range in bullying victimization rates of students in different secondary schools in this study was from 6.8% to 37.3%, with a median of 18.4% (12.5%, 22.4%). Multiple bullying victimization rates ranged from 0.9% to 14.8%, with a median of 6.1% (4.0%, 8.8%).

### 3.3. Results of the Multilevel Model Analysis of Bullying Victimization

The results of the null model (Model 1) showed that 5.1% of the total variation in bullying victimization could be explained by differences between schools. Further inclusion of pupil-level characteristics adjusted the models, and after adjusting for demographic characteristics (Model 2), physical and mental health characteristics (Model 3), and lifestyle characteristics (Model 4), a progressively lower proportion of the total variance in bullying victimization was explained at the school level, with the school-level variance being statistically significant in all models (*p* < 0.001).

In the full model (Model 5), boys (OR = 1.53, 95%CI: 1.43–1.65) aged less than 15 (OR = 1.59, 95%CI: 1.40–1.79), mothers with lower education (OR = 1.15, 95%CI: 1.00–1.32), with depressive symptoms (OR = 3.61, 95%CI: 3.34–3.90), overweight and obesity (OR = 1.28, 95%CI: 1.19–1.38), and smoking (OR = 1.31, 95%CI: 1.15–1.48), alcohol consumption (OR = 1.41, 95%CI: 1.31–1.53), addictive drug use (OR = 1.26, 95%CI: 1.17–1.36), intake of sugary drinks (OR = 1.20, 95%CI: 1.09–1.32), watching television (OR = 1.11, 95%CI: 1.02–1.21), and lack of sleep (OR = 1.37, 95%CI: 1.23–1.52) increased the OR of bullying victimization among students. The OR of bullying victimization was lower for students living in nuclear families (OR = 0.93, 95%CI: 0.86–0.99) and in schools with an established bullying disposal process (OR = 0.83, 95%CI: 0.71–0.99). See Table 2 for details.

### 3.4. Results of the Multilevel Model Analysis of Multiple Bullying Victimization

The results of the null model (Model 1) showed that 5.8% of the total variance in multiple bullying victimization rates could be explained by differences between schools. Further inclusion of pupil-level characteristics adjusted the models, and after adjusting for demographic characteristics (Model 2), physical and mental health characteristics (Model 3), and lifestyle characteristics (Model 4), the proportion of the total variance in multiple bullying victimization explained at the school level decreased progressively, with statistically significant school-level variance in all models (*p* < 0.001).

In the full model (Model 5), boys (OR = 1.72, 95%CI: 1.54–1.93) aged less than 15 years (OR = 1.55, the 95%CI: 1.32–1.82) with depressive symptoms (OR = 4.98, 95%CI: 4.45–5.57), overweight and obesity (OR = 1.23, 95% CI: 1.10–1.37), smoking (OR = 1.42, 95%CI: 1.20–1.68), alcohol consumption (OR = 1.26, 95%CI: 1.11–1.42), addictive drug use (OR = 1.36, 95%CI: 1.21–1.51), and lack of sleep (OR = 1.33, 95%CI: 1.14–1.55) had an increased OR of multiple bullying victimization for students. None of the correlations between school-level characteristics and multiple bullying victimization were statistically significant. See Table 3 for details.

## 4. Discussion

This study represents the first empirical investigation in mainland China to explore the association between bullying prevention and control measures and the incidence of bullying victimization, including multiple instances, within school settings. This study found that, after controlling for demographic factors, physical and mental health indicators, and lifestyle characteristics of secondary school students, the implementation of a bullying disposal process within schools was associated with a decreased likelihood of bullying victimization among secondary school students. However, this association did not reach statistical significance for multiple instances of bullying victimization. This study further revealed that the establishment of a bullying governance committee, thematic education for students, thematic training for parents, and a bullying-specific survey did not demonstrate a significant association with student bullying victimization or multiple instances of bullying victimization. Moreover, this study identified several factors that were significantly associated with an increased percentage of bullying victimization. These factors include being boys, being under 15 years old, having a mother with lower educational attainment, experiencing depressive symptoms, being overweight or obese, engaging in smoking, alcohol consumption, addictive drug use, frequent intake of sugary drinks, excessive television watching, and insufficient sleep. Conversely, belonging to a nuclear family was significantly associated with a decreased percentage of bullying victimization. At the same time, boys under 15 years of age, experiencing depression symptoms, being overweight or obese, smoking, alcohol consumption, addictive drug use, and lack of sleep were significantly associated with increased rates of multiple bullying victimization.

Firstly, this study found that the establishment of a bullying disposal process was strongly associated with lower rates of bullying victimization. This finding aligns with prior research that has consistently reported the positive impact of implementing school-based bullying interventions and processes in identifying and mitigating the risk of bullying victimization [30,31]. Insufficient bullying prevention mechanisms and a lack of educators’ awareness regarding student bullying protection emerged as crucial factors contributing to the occurrence of student bullying [32]. A study from Norway found that the implementation of the O’s bullying prevention program in schools resulted in increased awareness, preparedness, and competence to deal with and prevent bullying, as well as a change in the pre-existing school culture and, in turn, a reduction in bullying behavior among students [33]. An intervention study from the United States in 2020 also showed that school bullying interventions were effective in reducing bullying victimization and that the effects of the intervention were sustained over a longer period of time [34]. The key to reducing bullying in schools lies in improving the school environment through methods including education and disposal [35]. It has also been documented that a comprehensive approach combining school-wide bullying prevention programs and targeted interventions for individual students was necessary to effectively mitigate the risk of bullying victimization [36]. In order to effectively prevent bullying in schools, China’s Ministry of Education issued the Work Programme for Special Control Actions to Prevent Bullying among Primary and Secondary School Students in 2021. The action program placed particular emphasis on the need to deal seriously with bullying in schools in accordance with the law, in particular by establishing a standardized bullying reporting system [37]. Therefore, it is crucial to establish appropriate bullying disposal procedures for addressing specific incidents as well as providing individual support. Additionally, reducing the rate of bullying victimization necessitates the implementation of a combination of targeted interventions for individual students and comprehensive whole-school programs.

Secondly, this study found no significant association between the establishment of a bullying disposal process and multiple bullying victimization. Similarly, this study did not find a significant association between the establishment of a bullying governance committee, thematic education for students, professional training for parents, and specialized investigations into bullying, with either bullying victimization or multiple bullying victimization. Previous research has found that the presence of school bullying prevention and control measures was associated with lower rates of physical, verbal, and relational bullying [31]. In addition, Woods and Wolke’s study found that students in schools that implemented high-quality bullying prevention and control measures had lower rates of physical, verbal, and property bullying victimization [38]. Furthermore, several studies have indicated that the currently implemented anti-bullying interventions have had limited effectiveness in reducing the risk of bullying victimization. For instance, interventions developed for individual students in Australian schools did not demonstrate statistical significance in their effects [39]. A cluster-randomized controlled trial conducted in UK secondary schools revealed that a curriculum focusing on social and emotional skills for students had minimal impact on reducing bullying. However, the study demonstrated that a comprehensive intervention approach appeared to be effective [40]. The potential reasons for the limited effectiveness of the interventions in this study included inadequate implementation of the relevant measures and a short duration of their implementation [41]. The extent to which schools implemented student bullying prevention and control measures could vary widely by region and policy content, which may affect the relationship between the existence and implementation of preventive measures and the incidence of student bullying. On the other hand, due to the relatively delayed implementation of bullying prevention and control measures in Chinese schools following the enactment of the national policy, the desired effects of these measures may not have been fully realized yet. In order to address the above issues, the Office of the Education Supervisory Commission of the State Council of China decided to carry out the Year of Implementation of Bullying Prevention and Control for Primary and Secondary School Students action in 2018, thereby implementing the Guiding Opinions on Preventing and Combating Bullying and Violence among Primary and Secondary School Students, which was released in 2017 [18]. While current research indicated no statistically significant association between bullying prevention measures and bullying victimization, it is crucial to ensure consistent implementation and ongoing monitoring of policies and measures.

Thirdly, our study found that boys under 15 years old, mothers with lower levels of education, depression, overweight and obesity, smoking, alcohol use, addictive drug use, intake of sugary beverages, television viewing, and lack of sleep were significantly associated with increased rates of bullying victimization. At the same time, boys under 15 years old, depression, overweight and obesity, smoking, drinking alcohol, addictive drug use, and lack of sleep were strongly associated with increased rates of multiple bullying victimization. This finding is consistent with existing research that has shown personal characteristics such as gender and age to be strongly associated with the risk of both bullying and multiple bullying victimization [42]. Moreover, lifestyle behaviors such as smoking, alcohol consumption, drug addiction, and sleep deprivation also demonstrated a strong association with the risk of bullying victimization and multiple bullying victimization [43,44]. Furthermore, our study found a strong association between the intake of sugary drinks and the risk of bullying among students. While previous research has indicated that the intake of sugary drinks contributes to school violence [45], our study serves as a valuable addition to the existing body of research.

This study has several limitations. First, due to the cross-sectional design, causal associations cannot be asserted, and therefore, the findings should be interpreted with caution. Future prospective studies are warranted to establish the causal relationship between different factors and bullying victimization, as well as to evaluate the effectiveness of bullying prevention measures implemented in schools. Second, it is worth mentioning that the completion of student questionnaires was self-reported, which may introduce some degree of recall bias. Third, it is important to consider that some victims of bullying may hesitate to disclose their experiences of victimization, potentially leading to an underestimation of the rate of student bullying victimization.

## 5. Conclusions

The research objective of this study was to explore the association between bullying prevention and control measures and the rates of bullying victimization and multiple bullying victimization among secondary school students in China. Following the utilization of multilevel logistic regression models to examine the relationship between bullying prevention and control measures in Chinese schools and bullying behavior among secondary school students, this study found that the implementation of a bullying disposal procedure was associated with a decrease in the rate of bullying victimization. However, this association was not observed with the rate of multiple instances of bullying victimization. Furthermore, our study revealed that the implementation of a bullying governance committee, thematic education for students, thematic training for parents, and a bullying-specific survey were not found to be significantly associated with the risk of bullying or multiple instances of bullying among Chinese secondary school students. The lack of effectiveness of bullying prevention and control measures in schools may be related to the lagging nature of the policy as well as the lack of implementation of the policy. As a result, education supervisory authorities should monitor the implementation of bullying prevention and control measures in schools and incorporate the effectiveness of bullying prevention and control measures into the school evaluation and assessment system. In summary, this study provides important empirical support for the formulation of bullying policies and the implementation of related measures in schools. It emphasizes the importance of garnering attention from families, schools, the government, and all relevant sectors of society regarding the formulation of effective bullying policies and the implementation of prevention and control measures. These efforts will undoubtedly contribute to safeguarding the physical and mental health and safety of Chinese students.

## Figures and Tables

**Figure 1 behavsci-13-00954-f001:**
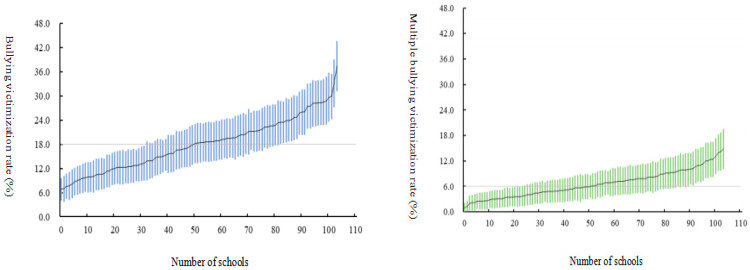
Prevalence of bullying victimization and multiple bullying victimization in different secondary schools.

**Table 1 behavsci-13-00954-t001:** Distribution of bullying victimization and multiple bullying victimization.

	Participants (%)	Victims of Bullying (%)	Victims of Multiple Bullying (%)
Gender			
Boys	13,276 (52.3%)	2777 (61.1%)	1064 (64.4%)
Girls	12,103 (47.7%)	1767 (38.9%)	587 (35.6%)
Age (year)			
<15	12,264 (48.3%)	2530 (55.7%)	911 (55.2%)
>=15	13,115 (51.7%)	2014 (42.3%)	740 (44.8%)
Region			
Urban	15,899 (62.6%)	2623 (57.7%)	924 (56.0%)
Rural	9480 (37.4%)	1921 (42.3%)	727 (44.0%)
Family structure			
Nuclear family	11,567 (45.6%)	1937 (42.6%)	683 (41.4%)
Non-nuclear family	13,812 (54.4%)	2607 (57.4%)	968 (58.6%)
Mother’s education level			
Primary school and below	3810 (15.0%)	835 (18.4%)	314 (19.0%)
Junior or senior high school	18,368 (72.4%)	3166 (69.7%)	1138 (68.9%)
University and above	3201 (12.6%)	543 (11.9%)	199 (12.1%)
Depressive symptom			
Yes	4801 (18.9%)	1779 (39.2%)	848 (51.4%)
No	20,578 (81.1%)	2765 (60.8%)	803 (48.6%)
Overweight and obesity			
Yes	8118 (32.0%)	1708 (37.6%)	630 (38.2%)
No	17,261 (68.0%)	2836 (62.4%)	1021 (61.8%)
Smoking			
Yes	1778 (7.0%)	558 (12.3%)	250 (15.1%)
No	23,601 (93.0%)	3986 (87.7%)	1401 (84.9%)
Alcohol consumption			
Yes	7090 (27.9%)	1779 (39.2%)	689 (41.7%)
No	18,289 (72.1%)	2765 (60.8%)	962 (58.3%)
Addictive drug use			
Yes	7557 (19.8%)	1680 (37.0%)	672 (40.7%)
No	17,822 (70.2%)	2864 (63.0%)	979 (59.3%)
Intake of sugary drinks			
Yes	20,454 (80.6%)	3828 (84.2%)	1370 (83.0%)
No	4925 (19.4%)	716 (15.8%)	281 (17.0%)
Intake of fruits			
Yes	24,829 (97.8%)	4413 (97.1%)	1584 (95.9%)
No	550 (2.2%)	131 (2.9%)	67 (4.1%)
Lack of physical activity			
Yes	3983 (15.7%)	781 (17.2%)	286 (17.3%)
No	21,396 (84.3%)	3763 (82.8%)	1365 (82.7%)
Television viewing			
Yes	17,518 (69.0%)	3263 (71.8%)	1171 (70.9%)
No	7861 (31.0%)	1281 (28.2%)	480 (29.1%)
Computer use			
Yes	14,363 (56.6%)	2582 (56.8%)	933 (56.5%)
No	11,016 (43.4%)	1962 (43.2%)	718 (43.5%)
Lack of sleep			
Yes	2616 (10.3%)	670 (14.7%)	272 (16.5%)
No	22,763 (89.7%)	3874 (85.3%)	1379 (83.5%)
Establishment of Bullying Governance Committee			
Yes	21,591 (85.1%)	3921 (86.3%)	1423 (86.2%)
No	3788 (14.9%)	623 (13.7%)	288 (13.8%)
Conducting thematic education for students			
Yes	23,690 (93.3%)	4256 (93.7%)	1541 (93.3%)
No	1689 (6.7%)	288 (6.3%)	110 (6.7%)
Conducting specialized training for parents			
Yes	21,364 (84.2%)	3792 (83.4%)	1361 (82.4%)
No	4015 (15.8%)	752 (16.6%)	290 (17.6%)
Conducting a special survey on bullying			
Yes	23,252 (91.6%)	4154 (91.4%)	1501 (90.9%)
No	2127 (8.4%)	390 (8.6%)	150 (9.1%)
Establishing a bullying disposal process			
Yes	16,791 (66.2%)	2817 (62.0%)	1019 (61.7%)
No	8588 (33.8%)	1727 (38.0%)	632 (38.3%)
Total	25,379 (100.0%)	4544 (100.0%)	1651 (100.0%)

**Table 2 behavsci-13-00954-t002:** Results of the multilevel model analysis of student bullying victimization.

Model	Model 1	Model 2	Model 3	Model 4	Model 5 (Core Model)
Random effects models					
School level variance	0.1757	0.1298	0.1249	0.1073	0.0960
ICC	0.0507	0.0403	0.0384	0.0331	0.0299
*p*-value	<0.001	<0.001	<0.001	<0.001	<0.001
MOR	1.49	1.41	1.40	1.37	1.34
PCV (%)		26.12	28.91	38.93	45.36
Fixed effects model					
Intercept	−1.5735	−1.9557	−2.4883	−2.8973	−2.8973
*p*-value	<0.001	<0.001	<0.001	<0.001	<0.001
Student-level characteristics					
Gender					
Boys		**1.54 (1.44–1.65)**	**1.63 (1.51–1.75)**	**1.53 (1.43–1.65)**	**1.53 (1.43–1.65)**
Girls		1.00	1.00	1.00	1.00
Age (years)					
<15		**1.35 (1.19–1.52)**	**1.48 (1.31–1.68)**	**1.60 (1.42–1.82)**	**1.59 (1.40–1.79)**
>=15		1.00	1.00	1.00	1.00
Region					
Urban		1.00	1.00	1.00	1.00
Rural		1.18 (1.00–1.38)	1.17 (1.00–1.38)	1.15 (0.99–1.35)	1.09 (0.94–1.27)
Family structures					
Nuclear family		**0.88 (0.83–0.95)**	**0.91 (0.85–0.98)**	**0.93 (0.86–1.00)**	**0.93 (0.86–0.99)**
Non-nuclear families		1.00	1.00	1.00	1.00
Mother’s education level					
Primary school and below		**1.20 (1.05–1.36)**	**1.16 (1.01–1.33)**	**1.15 (1.00–1.32)**	**1.15 (1.00–1.32)**
Junior or senior high school		0.95 (0.85–1.06)	0.95 (0.85–1.06)	0.95 (0.85–1.06)	0.95 (0.85–1.06)
University and above		1.00	1.00	1.00	1.00
Depressive symptom					
Yes			**4.14 (3.83–4.46)**	**3.61 (3.34–3.90)**	**3.61 (3.34–3.90)**
No					
Overweight and obesity					
Yes			**1.29 (1.20–1.38)**	**1.28 (1.19–1.38)**	**1.28 (1.19–1.38)**
No			1.00	1.00	1.00
Smoking					
Yes				**1.31 (1.16–1.48)**	**1.31 (1.15–1.48)**
No				1.00	1.00
Drinking alcohol					
Yes				**1.41 (1.30–1.53)**	**1.41 (1.30–1.53)**
No				1.00	1.00
Addictive drug use					
Yes				**1.27 (1.18–1.36)**	**1.26 (1.17–1.36)**
No				1.00	1.00
Intake of sugary drinks					
Yes				**1.19 (1.09–1.31)**	**1.20 (1.09–1.32)**
No				1.00	1.00
Intake of fruits					
Yes				1.00	1.00
No				1.11 (0.89–1.38)	1.11 (0.89–1.39)
Lack of physical activity					
Yes				1.04 (0.94–1.14)	1.04 (0.95–1.14)
No				1.00	1.00
Television viewing					
Yes				**1.12 (1.03–1.21)**	**1.11 (1.02–1.21)**
No				1.00	1.00
Computer use					
Yes				0.94 (0.87–1.01)	0.94 (0.87–1.01)
No				1.00	1.00
Lack of sleep					
Yes				**1.36 (1.22–1.52)**	**1.37 (1.23–1.52)**
No				1.00	1.00
Establishment of Bullying Governance Committee					
Yes					1.30 (0.98–1.72)
No					1.00
Conducting thematic education for students					
Yes					1.27 (0.70–2.29)
No					1.00
Conducting specialized training for parents					
Yes					0.81 (0.62–1.04)
No					1.00
Conducting a special survey on bullying					
Yes					0.88 (0.52–1.48)
No					1.00
Establishing a bullying disposal process					
Yes					**0.83 (0.71–0.99)**
No					1.00

Note: Bold indicates statistically significant differences.

**Table 3 behavsci-13-00954-t003:** Results of the multilevel model analysis of multiple bullying victimization of students.

Model	Model 1	Model 2	Model 3	Model 4	Model 5 (Core Model)
Random effects models					
School-level variance	0.2037	0.1578	0.1376	0.1182	0.1068
ICC	0.0583	0.0484	0.0418	0.0361	0.0329
*p*-value	<0.001	<0.001	<0.001	<0.001	<0.001
MOR	1.54	1.46	1.42	1.39	1.36
PCV (%)		22.53	32.45	41.97	47.57
Fixed effects model					
Intercept	−2.7447	−3.1491	−2.4883	−4.1395	−4.1070
*p*-value	<0.001	<0.001	<0.001	<0.001	<0.001
Student-level characteristics					
Gender					
Boys		**1.69 (1.52–1.88)**	**1.83 (1.64–2.05)**	**1.72 (1.54–1.93)**	**1.72 (1.54–1.93)**
Girls		1.00	1.00	1.00	1.00
Age (years)					
<15		**1.29 (1.09–1.52)**	**1.46 (1.24–1.73)**	**1.57 (1.33–1.85)**	**1.55 (1.32–1.82)**
>=15		1.00	1.00	1.00	1.00
Region					
Urban		1.00	1.00	1.00	1.00
Rural		**1.24 (1.02–1.51)**	**1.22 (1.01–1.47)**	1.19 (1.00–1.43)	1.15 (0.95–1.38)
Family structures					
Nuclear family		**0.85 (0.76–0.94)**	**0.88 (0.79–0.99)**	0.90 (0.81–1.00)	0.90 (0.81–1.00)
Non-nuclear families		1.00	1.00	1.00	1.00
Mother’s education level					
Primary school and below		1.17 (0.96–1.43)	1.11 (0.90–1.36)	1.11 (0.90–1.36)	1.11 (0.91–1.37)
Junior or senior high school		0.92 (0.78–1.09)	0.95 (0.85–1.06)	0.93 (0.79–1.10)	0.93 (0.79–1.10)
University and above		1.00	1.00	1.00	1.00
Depressive symptom					
Yes			**5.68 (5.10–6.32)**	**4.97 (4.45–5.56)**	**4.98 (4.45–5.57)**
No					
Overweight and obesity					
Yes			**1.23 (1.10–1.37)**	**1.23 (1.10–1.37)**	**1.23 (1.10–1.37)**
No			1.00	1.00	1.00
Smoking					
Yes				**1.43 (1.20–1.69)**	**1.42 (1.20–1.68)**
No				1.00	1.00
Drinking alcohol					
Yes				**1.26 (1.11–1.42)**	**1.26 (1.11–1.42)**
No				1.00	1.00
Addictive drug use					
Yes				**1.36 (1.22–1.52)**	**1.36 (1.21–1.51)**
No				1.00	1.00
Intake of sugary drinks					
Yes				1.03 (0.89–1.19)	1.03 (0.89–1.19)
No				1.00	1.00
Intake of fruits					
Yes				1.00	1.00
No				1.46 (1.09–1.94)	1.46 (1.09–1.95)
Lack of physical activity					
Yes				0.97 (0.84–1.12)	0.96 (0.83–1.11)
No				1.00	1.00
Watching television					
Yes				1.07 (0.94–1.22)	1.07 (0.94–1.22)
No				1.00	1.00
Computer use					
Yes				0.93 (0.83–1.05)	0.93 (0.83–1.05)
No				1.00	1.00
Lack of sleep					
Yes				**1.32 (1.14–1.54)**	**1.33 (1.14–1.55)**
No				1.00	1.00
Establishment of Bullying Governance Committee					
Yes					1.30 (0.93–1.82)
No					1.00
Conducting thematic education for students					
Yes					1.32 (0.66–2.65)
No					1.00
Conducting specialized training for parents					
Yes					0.75 (0.55–1.01)
No					1.00
Conducting a special survey on bullying					
Yes					0.83 (0.45–1.52)
No					1.00
Establishing a bullying disposal process					
Yes					0.89 (0.73–1.09)
No					1.00

Note: Bold indicates that the difference is statistically significant.

## Data Availability

The data, methods, and materials used in this study can be obtained by contacting the corresponding author.

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
