# Peer review of "An Analysis of the Association between School Bullying Prevention and Control Measures and Secondary School Students’ Bullying Behavior in Jiangsu Province"

_behavsci, 2023, doi:10.3390/bs13110954_

Round 1

Reviewer 1 Report

Comments and Suggestions for Authors

This is an interesting and a significant study because school bullying is a global problem. The study shows the current state, not, in my opinion, the functionality of the measures. The functionality of the measure cannot be determined without a prior survey of the state without measures. The results of anti-bullying measures would be shown in longitudinal research - that is, survey of the situation, implementation of measures, follow-up survey. Anti-bullying measures include establishing an anti-bullying committee (who is on the committee?), student education, parent training, research on bullying. Why is training for educators not mentioned as a measure? Educators should be able to recognize possible bullying in time and implement measures.

I recommend:

add who is on the bullying detection committee

whether there are trainings for educators or why they are not

Author Response

Dear reviewer:

Thank you for your valuable comments on this manuscript. We have carefully revised the manuscript according to your comments. All revisions have been highlighted in red in the newly submitted manuscript. Please see our point by point responses listed hereunder. Please see the attachment.

Reviewer 2 Report

Comments and Suggestions for Authors

First of all, I would like to thank you for the opportunity to read your work. I found this study original because of the bullying prevention proposal. Please find below specific comments and suggestions for your consideration:

1. The title of the article: An Analysis of the Association Between School Bullying Prevention and Control Measures and Secondary School Students' Bullying Behavior. It should be conducted with in Jiangsu Province (China).

2. You should strengthen your study by stating the research objective, variables or hypotheses and introducing bibliographical references.

3. They should provide a description of the methodology and objective of the research in the abstract. 

4. They should explain the degree of representativeness of the sample. In other words, describe the total population of the study and its representativeness.

5. In the discussion they should talk about what measures or studies have been published since the study they conducted in 2019.

6. They should expand on the conclusions and relate them to the objective(s) of the study.

Author Response

(The authors gave the same response as above.)

Reviewer 3 Report

Comments and Suggestions for Authors

Thank you for the opportunity for me to review your manuscripts with behavioral science-Behavioral Sciences. Your topic is worthy. However, I have some concerns for you and I hope they are helpful.

1.     Abstract. The abstract is vague and not clear. It needs improvement thoroughly. For example, the background is too long and not clear to address the needs in the fields. You just said that there is a lack of empirical research but no evidence. Also, the method part is not clear and needs improvement. The multilevel logistic regression models are used in this study. Overall, it needs further improvement and makes it more accurate.

2.    Introduction. Although the introduction indicates the fields and needs of this topic, the narrative needs improvement and proofreading. Also, some sentences are confusing. I suggest that do not use many long sentences. Thank you.

3.    Materials and methods. I wonder about the measure parts and the data criteria and variables. They are not clear. You use four models and what are your models?

4.    Results. Overall, there are good tables for presentation. However, the results need improvement and thorough revision. For example, what are the regression coefficients and variance components for the multilevel logistic regression models and odd ratio? Also, some terms need to be changed and revised such as “gen” and “modeling” and so forth. The major issue is that this manuscript has no research questions, which confuse readers. Therefore, this manuscript needs improvement.

5.     The discussion and conclusion need improvement and proofreading. Also, the study needs more recent data to support this study. 

Overall, you did a great job here, and good luck.

Comments on the Quality of English Language

Proofreading is needed. 

Author Response

(The authors gave the same response as above.)

Round 2

Reviewer 2 Report

Comments and Suggestions for Authors

The article has been greatly improved and should be published. 

Reviewer 3 Report

Comments and Suggestions for Authors

To the authors, 

Thank you for giving me this opportunity to re-review your manuscript, which is well done. 

Thank you, and good luck. 

Comments on the Quality of English Language

No comments. I suggest if there has a proofreading, which is better.